# Transcriptome Analysis of Marbled Rockfish *Sebastiscus marmoratus* under Salinity Stress

**DOI:** 10.3390/ani13030400

**Published:** 2023-01-24

**Authors:** Zhiqi He, Chenyan Shou, Zhiqiang Han

**Affiliations:** Fishery College, Zhejiang Ocean University, Zhoushan 316002, China

**Keywords:** *Sebastiscus marmoratus*, salinity stress, RNA-seq, differentially expressed genes

## Abstract

**Simple Summary:**

Changes in salinity can have an impact on the physiological processes of fish. In this study, RNA-seq sequencing was used to detect and analyze the differential gene expression of *Sebastiscus marmoratus* in salinity fluctuations and to explore the genes and pathways related to salinity stress. The results demonstrated that pik3r6b, cPLA2γ-like, and WSB1 were differentially expressed in all three groups. All three genes were highly expressed in the high-salt group, with wsb1 being more expressed than pik3r6b and cPLA2γ-like. The results of this study provide some basic theoretical references for the salinity adaptation mechanism of *Sebastiscus marmoratus*.

**Abstract:**

The marbled rockfish, *Sebastiscus marmoratus*, belongs to the euryhaline fish and is an oviparous scleractinian fish. There are few studies on the adaptation mechanism, functional genes, and related pathways of *S. marmoratus* and salinity. The results showed that a total of 72.1 GB of clean reads were obtained and all clean reads annotated a total of 25,278 Unigenes, of which 2,160 were novel genes. Compared to 20‰, 479 and 520 differential genes were obtained for 35‰ and 10‰, respectively. Gene ontology (GO) enrichment analysis revealed significant enrichment in protein binding, ion binding, ATP binding, and catalytic activity. Kyoto Encyclopedia of Genes and Genomes (KEGG) showed that differentially expressed genes significantly expressed under salinity stress were mainly involved in the pathways of the cytochrome P450 metabolism of xenobiotics, tryptophan metabolism, cellular senescence, and calcium signaling pathways. Among them, pik3r6b, cPLA2γ-like, and WSB1 were differentially expressed in all three groups, and they were associated with apoptosis, inflammation, DNA damage, immune regulation, and other physiological processes. Six differentially expressed genes were randomly selected for qRT-PCR validation, and the results showed that the transcriptomic data were of high confidence.

## 1. Introduction

Salinity is one of the environmental factors on which all aquatic organisms depend, and today’s increasingly harsh climate change is already having a range of adverse effects on marine ecosystems [1,2]. Changes in freshwater input patterns, ocean circulation, and precipitation are due to global warming, which can cause changes in seawater salinity [3,4].

Salinity fluctuations can directly affect the physiological activities and distribution of fish and other aquatic organisms, in addition to adversely affecting aquaculture, infrastructure, and coastal ecosystems [5,6]. Previous studies have shown that most marine organisms may differ in their adaptation to salinity because their internal physiological environment is affected by changes in salinity [7]. Changes in salinity affect the hormonal control, energy metabolism, and immune processes of marine organisms due to the effect of changes in salinity on their osmoregulation [8]. This includes effects on fish growth and development [9], metabolism, and osmotic pressure regulation [10], as well as nutrient requirements and physiological and biochemical indicators [11]. In addition, there are some differences in adaptation to salinity changes between species, and the same species has different responses to salinity stress, and this variability may be related to the osmotic capacity of the species [12,13,14]. Most Osteichthyes are capable of osmoregulation, which they use to maintain internal osmotic pressure, and their adaptation to salinity stress is based on effective mechanisms of osmotic sensing and osmotic stress signaling, with fluctuations in salinity causing hypertonic or hypotonic stress in fish [15,16]. This has been reported in many cases, for example, in *Oreochromis niloticus* [17], *Nibea japonica* [18], *Rachycentron canadum* [19], *Paralichthys olivaceus* [20], and *Charybdis japonica* [21]. In summary, many studies have reported on the effects of salinity fluctuations on fish but less on the molecular regulatory mechanisms. Therefore, it is necessary to investigate the regulatory mechanisms of *S. marmoratus* under salinity fluctuations at the molecular level.

The marbled rockfish, *Sebastiscus marmoratus*, belongs to the euryhaline fish and is an oviparous Osteichthyes fish. *S. marmoratus* belongs to the order Scorpaeniformes, family Scorpaenidae, genus *Sebastiscus*, and it is found off the coast of China, in the western Pacific Ocean, and the inshore waters of the southern coast. It is an economically important fish species for aquaculture [22,23]. The salinity range of *S. marmoratus* is 10–34, and the optimum salinity range is 16–22. When the salinity is lower than 10 or higher than 34, it will affect the survival of *S. marmoratus* and can lead to death [24]. Transcriptome sequencing not only helps map and annotate the fish transcriptome but also facilitates the understanding of biological processes, such as growth and development, adaptive evolution, and immune and stress responses in fish [25]. RNA-seq techniques can provide insights into the adaptation mechanisms of aquatic animals to salinity by mining for functionally expressed genes under fluctuating salinity [26]. Many scholars have used RNA-seq to analyze the salinity adaptation mechanisms of aquatic animals and uncover salinity-related genes or pathways. For example, *Scophthalmus maximus* [27], *Acipenser baeri* [28], *Oratosquilla oratoria* [29], and *Argyrosomus japonicus* [30]. Currently, studies on S. marmoratus have focused on macroscopic aspects, such as population genetics [31] and growth and development [32], and less on the mechanisms of adaptation to salinity, functional genes, and related pathways.

In this study, the transcriptome of the gill tissue of *S. marmoratus* was sequenced under different salinity gradients. This tissue was chosen because it is one of the osmoregulatory organs of fish [33]. We identified and analyzed the differential gene expression in *S. marmoratus* under salinity fluctuations and explored the functional genes and pathways associated with salinity stress. Thus, elucidating these scenarios will help us to understand the regulatory mechanisms of *S. marmoratus* under salinity fluctuations and provide data and a theoretical basis for the adaptive mechanisms of *S. marmoratus* in salinity stress environments.

## 2. Materials and Methods

### 2.1. Experimental Materials and Salinity Stress Experiment

In this salinity stress experiment, a total of 30 *S. marmoratus* of similar body sizes were randomly collected in the offshore waters of Zhoushan, China. A total of 30 *S. marmoratus* were temporarily kept in plastic water tanks (55 cm × 40 cm × 35 cm) for 48 h before the start of the formal experiment, during which the water temperature was 20 °C and salinity 20‰, feeding was stopped and oxygen was continuously supplied. After 48 h, 18 *S. marmoratus* were randomly selected and exposed to different salinity gradients for 12 h. All 18 *S. marmoratus* were divided into 3 groups and placed in 3 plastic water tanks with 6 *S. marmoratus* in each group. The three salinity gradients were 10‰, 20‰, and 35‰, and 20‰ was set as the control group (Smsc); 10‰ and 35‰ were set as the low-salt group (Smsl) and high-salt group (Smsh), respectively, and the water temperature of each group was kept at 20 °C. At the end of the duress experiment, three *S. marmoratus* in good condition were randomly selected from each group, S. marmoratus was first frost-anesthetized, and finally gill tissue was collected from nine individuals. After sampling, the nine samples obtained were put into lyophilized tubes with RNA preservation solution, and then, the lyophilized tubes were numbered and snap frozen in liquid nitrogen and finally stored in a −80 °C refrigerator.

### 2.2. Total RNA Extraction, Library Construction, and High-Throughput Sequencing

After 9 samples were removed from the ultra-low-temperature refrigerator, total RNA was extracted from each sample using a standard TRIzol Reagent kit (Invitrogen, Carlsbad, CA, United States), and the RNA samples were then subjected to rigorous quality control. The RNA samples were analyzed by 1% agar gel electrophoresis for integrity and DNA contamination and then analyzed for RNA purity (OD260/280 and OD260/230 ratios) using a NanoPhotometer spectrophotometer (Invitrogen, Carlsbad, CA, United States). The Agilent 2100 bioanalyzer (Agilent Technologies, Santa Clara, CA, United States) was used to determine the integrity of the total RNA.

Library creation was performed using the NEBNext Ultra RNA Library Preparation kit (Illumina, San Diego, CA, United States). Firstly, the first strand of cDNA was synthesized in the M-MuLV reverse transcriptase system using fragmented mRNA as a template and random oligonucleotide as a primer. Then, the RNA strand was degraded by RNaseH, and the second strand was synthesized by dNTPs in the DNA polymerase I system. After purification, double-stranded cDNA was repaired at the end, A-tail was added, and the sequencing adaptor was connected. AMPure XP beads (Beckman Coulter, Beverly, United States) were used to screen 250–300 bp cDNA. PCR amplification was performed, and AMPure XP beads were used to purify the PCR products to obtain the final library. Finally, the libraries were sequenced on an Illumina HiSeq 2000 with 150 bp paired ends.

### 2.3. Reference Genome Alignment and De Novo Transcriptome Transfer, Annotation

Using the S. marmoratus genome (https://doi.org/10.6084/m9.figshare.19161290.v2, accessed on 27 May 2022) as the reference genome, the clean reads after quality control were compared with the reference genome, and the raw data obtained from sequencing were quality controlled, mainly removing reads with connectors, removing the N-containing reads, removing low-quality reads, and obtaining clean reads after quality control. New transcripts were made using StingTie software. After assembling, the transcripts were annotated with Pfam (Protein family database), SUPERFAMILY database, GO (Gene Ontology), KEGG (Kyoto Encyclopedia of Genes and Genomes), and other databases.

### 2.4. Quantitative Analysis of Differential Expression Levels and Differential Gene Enrichment Analysis

The expression of genes was calculated using the fragments per kilobase of exon model per million mapped fragments (FPKM) method, and the expression levels of genes were calculated by comparing the number of clean reads to the reference gene set. After the quantitative analysis of gene expression levels was completed, the differential expression analysis of each experimental group was then performed using DEseq2, and FDR < 0.05 and |log2FC| >1 were used as the conditions for screening differentially expressed genes, and then, finally, GOseq software with KOBAS (2.0) was used for GO functional annotation and KEGG pathway enrichment analysis of differential genes.

### 2.5. RT-qPCR Analysis

Six differentially expressed genes (DEGs) were randomly selected to validate the RNA-Seq results, and primers were designed with Primer Premier 5.0 using β-actin as the internal reference gene, and the primers are shown in Table 1. The qRT-PCR was designed according to the TB Green^®^ Premix Ex TaqTM (Tli RNase H Plus, Osaka, Japan) RR420A manufacturer’s instructions, and the real-time quantitative PCR instrument was used with an ABI 7300 Plus real-time PCR instrument (Applied Biosystems, Carlsbad, CA, USA). The cDNA was used as the template, and the dilution of cDNA was determined using the standard curve. The reaction system was as follows: 2 × TB Green Premix Ex Taq (Tli RNase H Plus) 10 μL, 50 × ROX Reference Dye II 0.4 μL, nuclease-free water 6.8 μL, 0.4 μL of each forward and reverse primer, cDNA sample 2 μL. The PCR reaction conditions were 95 °C for 30 s, 95 °C for 5 s, 60 °C for 30 s, 40 cycles, and melting curve analysis was performed after the reaction was completed. Three parallel experiments were performed for each cDNA template, and finally, the relative expression of genes was calculated using the 2^−ΔΔCt^ (ΔCT = CTtarget gene − CTreference gene, ΔΔCT = ΔCTtreatment − ΔCTcontrol) method [34]. The data were processed using SPSS 19.0, and one-way analysis of variance (ANOVA) was used to compare the differences between groups, with *p* < 0.05 as the threshold of significant difference.

## 3. Results

### 3.1. Transcriptome Sequencing Data and Reference Genome Alignment Analysis

We obtained raw reads from nine samples of *S. marmoratus* subjected to different salinity treatments by Illumina-based RNA sequencing and deposited them into the National Center for Biotechnology Information database under accession number SRR23019908 to SRR23019916 under BioProject PRJNA921893 and BioSample SAMN32638115. RNA-Seq was performed on gill tissues from the *S. marmoratus* high-salt group (Smsh), control group (Smsc), and low-salt group (Smsl), and the raw data were obtained, and the clean reads after quality control are shown in Table 2. The results showed that a total of 486,746,342 raw reads were obtained, 480,746,744 clean reads were obtained after removing low-quality sequences, and the filtered clean data of each sample exceeded 6.86 GB. The percentage of Q30 bases was not less than 91.99%, and the GC content ranged from 48.62% to 49.27%. The clean reads of each sample compared with the sequences of the reference genome using Hista2 software (Table 3) showed that the matches ranged from 86.31% to 93.19%, indicating a high-confidence level of the transcriptome sequencing results.

### 3.2. Differential Gene Expression

Gene expression was calculated for each sample according to the expected number of fragments per kilobase of transcript sequence per million base pairs sequenced (FPKM) method, and the expression level of each sample was represented by a box plot in order to examine the distribution of FPKM under different experimental conditions at an overall level. The box plot of gene expression (Figure 1a) showed consistent changes in gene expression levels for Smsh (35‰), Smsc (20‰), and Smsl (10‰), indicating that the transcriptome sequencing results were good for subsequent analysis. The sample correlation heat map of gene expression (Figure 1b) showed that the similarity of expression patterns between samples was high, indicating high experimental reliability and reasonable sample selection.

The differential genes in the high-salt group (35‰), control group (20‰), and low-salt group (10‰) were counted quantitatively, as shown in the volcano plot (Figure 2). There were 479 DEGs in the high-salt group, of which 176 DEGs indicated up-regulation and 303 DEGs indicated down-regulation; 520 DEGs in the low-salt group, of which 199 DEGs indicated up-regulation and 321 DEGs indicated down-regulation; and 233 DEGs in the high-salt vs. low-salt group, of which 152 were up-regulated genes and 81 were down-regulated genes.

As shown by the Venn diagram (Figure 3), there were 68 significantly differentially expressed genes in Smsh compared with Smsl, 82 and 40 differentially expressed genes in Smsh and Smsl compared with Smsh vs. Smsl. In addition, a total of three differential genes were significantly expressed in the three groups, namely, phosphatidylinositol 3-kinase regulatory subunit 6b (pik3r6b), cytoplasmic phospholipase A2-γ-like gene (cPLA2γ-like), and WD repeat and SOCS box containing 1 (WSB1). As shown in Table 4, the expression of their differential genes also differed according to salinity stress, with pik3r6b, cPLA2γ-like, and wsb1 all significantly differentially expressed in the high-salt group.

### 3.3. GO Functional Enrichment of Differentially Expressed Genes

GO enrichment analysis allows the identification of the main biological functions of DEGs as well as their properties. The differential genes were annotated in GO into three main categories: biological processes, cellular components, and molecular functions. Significant DEGs in the high-salt group (Figure 4a) involved organic metabolic processes (GO:0071704), macromolecule modification (GO:0043412), protein phosphorylation (GO:0006468), and other GO terms in the biological process. DEGs under cellular components involve GO terms such as membrane protein complex (GO:0098796), protein-containing complex (GO:0032991), and membrane part (GO:0044425); molecular functions involve GO terms such as ATP binding (GO:0005524), adenyl nucleotide binding (GO:0030554), adenyl ribonucleotide binding (GO:0032559), and other GO terms.

High-salt vs. low-salt group (Figure 4b), under biological process, involve carbohydrate derivative biosynthetic process (GO:1901137), metabolic process (GO:0008152), and primary metabolic process (GO:0044238); in cellular components, GO terms such as membrane part (GO:0044425), catalytic complex (GO:1902494), and membrane protein complex (GO:0098796) are involved; and in molecular functions, GO terms among the molecular functions involved transferase activity (GO:0016740), metallopeptidase activity (GO:0008237), transferase activity, transferring one-carbon groups (GO:0016741), and other GO terms.

The low-salt group (Figure 4c) is involved in biological processes such as response to stress (GO:0006950), organelle organization (GO:0006996), small GTPase-mediated signal transduction (GO: 0007264); in cellular components, GO terms such as actin cytoskeleton (GO:0015629), catalytic complex (GO:1902494), and membrane protein complex (GO:0098796) are involved; and GO terms such as motor activity (GO:0003774), carbohydrate derivative binding (GO:0097367), and anion binding (GO:0043168) are involved under molecular function.

### 3.4. KEGG Pathway Analysis of Differentially Expressed Genes

As shown by the KEGG (Kyoto Encyclopedia of Genes and Genomes) annotation analysis (Figure 5), the top 20 pathway annotation sequences showing significant enrichment in the high-salt group were mostly related to metabolic pathways, including nucleotide metabolism, biodegradation, and metabolism of xenobiotics, lipid metabolism, energy metabolism, amino acid metabolism, carbohydrate metabolism, and metabolism of cofactors and vitamins, and others were signaling molecules and interactions, membrane transport, cell growth and death, and the endocrine system. The annotated sequences of the top 20 pathways significantly enriched in the low-salt group were mostly related to biodegradation and metabolism of xenobiotics, carbohydrate metabolism, nucleotide metabolism, cofactor and vitamin metabolism, cell growth and death, signal transduction, signaling molecules and interactions, eukaryotic celluar community, and the circulatory system. The first 20 annotated sequences of pathways in the high-salt group compared with the low-salt group were mostly related to nucleotide metabolism, lipid metabolism, carbohydrate metabolism, cell growth and death, signaling molecules and interactions, and cofactor and vitamin metabolism. The annotated sequences in all three groups were significantly enriched in metabolic pathways, cell growth and death, and signaling molecules and interactions.

Using a *p*-value < 0.05 as a screening condition, significant DEGs involved 258 pathways. As shown in Figure 6, the most significantly enriched pathways in the high-salt group were the metabolism of xenobiotics by cytochrome P450 (ko00980; *p* = 4.20e−5) and drug metabolism - other enzymes (ko00983; *p* = 0.000296), which both pathways were up-regulated, with down-regulated genes enriched in ribosome biogenesis in eukaryotes (ko03008; *p* = 2.54e−35) and RNA polymerase (ko03020; *p* = 0.000877). The most significantly enriched pathway in the low-salt group was ECM-receptor interaction (ko04512; *p* = 7.87e−5), with up-regulated genes enriched in DNA replication (ko03030; *p* = 9.92e−5), tryptophan metabolism (ko00380; *p* = 0.0001364), and cellular senescence (ko04218; *p* = 0.0008194); down-regulated genes were enriched in ECM-receptor interaction, focal adhesion (ko04510; *p* = 0.000199), and calcium signaling pathway (ko04020; *p* = 0.001684). In the high-salt vs. low-salt group, there was no significant enrichment of up-regulated genes in the pathway, while down-regulated genes were mainly enriched in ribosome biogenesis in eukaryotes (ko03008; *p* = 1.02e−5) pathway.

### 3.5. Real-Time Quantitative PCR Results

To verify the accuracy of the transcriptome data, six DEGs were selected for qRT-PCR validation (Figure 7). The curves are based on the transcriptome results, and the bars are based on the qRT-PCR results. The results show that the expression trends of qRT-PCR results are consistent with those of RNA-Seq, indicating the reliability of the transcriptome sequencing results. In addition, the expression trends of different genes at different salinities were specific, either based on transcriptome or qRT-PCR, where the gene expression of pik3r6b, novel.1593, cyp1a1, and fgf10a increased with increasing or decreasing salinity, and the gene expression of dlgap1-like decreased with increasing or decreasing salinity. The expression of cPLA2γ-like genes increased with increasing salinity and decreased with decreasing salinity.

## 4. Discussion

Climate variability has resulted in constant changes in ocean salinity and temperature, and salinity stress has caused significant mortality of coastal marine organisms and coral reefs [35,36]. The euryhaline fish are highly adaptive to salinity, and their osmoregulatory capacity can quickly adapt to salinity changes caused by various situations [37]; the adaptability of fish to salinity changes is due to long-term selective evolution under environmental changes and is also a specific expression of genetic information [9]. Salinity changes cause fish to dynamically regulate the osmotic pressure inside and outside the organism and cause changes in their feeding, metabolism, and enzymes, finally causing disturbances in the internal environment with abnormalities in physiological indicators [18].

In saline environments, fish develop cellular stress responses in response to salinity stress, with repair and protection of cellular macromolecules, cell cycle arrest, and programmed cell death as fundamental aspects of the cellular stress response [15]. Marine Osteichthyes also actively secrete salts and retain water to maintain osmotic balance in the organism [16]. When fish are exposed to salinity stress, they use osmosensors to signal and induce large-scale molecular cascade responses and cellular remodeling to adjust or even completely switch from a high to a low osmolar physiological environment to restore endocytosis under widely fluctuating environmental salinity [3]. It has been shown that changes in seawater salinity directly affect the osmoregulatory mechanisms of aquatic animals [38] and that osmotic imbalances caused by salinity changes can impair metabolic, immune, and other physiological processes in aquatic animals [29]. However, so far no studies have shown the adaptation mechanisms of S. marmoratus to salinity and how salinity changes affect these physiological processes. Therefore, in this study, *S. marmoratus* was used as a study target to investigate the mechanisms of its response to salinity fluctuations.

In this study, *S. marmoratus* was exposed to a water environment in the high-salt group (35‰) vs. the low-salt group (10‰) and sequenced using the Illumina Hiseq platform. The results showed that the expression levels of many differential genes changed significantly with fluctuations in salinity, suggesting that many Unigenes are co-expressed under salinity stress and, thus, respond to salinity regulation mechanisms. When exposed to salinity stress, fish need to mobilize different numbers of genes to adapt to changes in the external environment to maintain their internal homeostasis. The number of differential genes in the low-salt group is greater than in the high-salt group, and similar gene expression trends have been found in *Charybdis japonica* [21]. Although gene expression changed significantly in the low-salt group, we cannot simply speculate that it was more stressed at that salinity. In addition, the number of up-regulated genes was higher in the low-salt group than in the high-salt group, presumably related to the salt tolerance mechanism of *S. marmoratus*.

In GO functional enrichment, DEGs are mainly enriched in GO terms such as metabolic processes, membranes, transmembrane transport, ATP binding, ion binding, and catalytic activity. This may be because the cellular function and structure of the *S. marmoratus* organism are altered when exposed to salinity stress, and many enzymes and proteins are stimulated in response to the salinity stimulus. Previous studies have shown that the cell membranes of fish gill tissue are in direct contact with the external water environment and are susceptible to changes in salinity [39], so fish maintain internal homeostasis by increasing ion pump activity in gill tissue or by reducing gill membrane permeability [40]. Most fish maintain osmoregulation by excreting salts and absorbing water [41]. In addition, Na+-K+-ATPase is the main ion transporter enzyme in aquatic animals, which responds to salinity adaptation mechanisms by enhancing osmoregulation [42]. Therefore, we suspect that *S. marmoratus* will strengthen its osmoregulation ability to adapt to the external environment under the change in salinity, and many key genes related to salinity will be activated or inhibited to maintain the balance of the body.

Most of the KEGG enrichment pathways are related to carbohydrate metabolism, cell growth and death, and signal transduction. Metabolism of xenobiotics by cytochrome P450, drug metabolism - other enzymes, two pathways associated with xenobiotic degradation and metabolism, and drug metabolism were significantly up-regulated in the high vs. low-salt groups. These two pathways are related to xenobiotics and metabolism and drug metabolism. Cytochrome P450 enzymes (CYP450) are drug-metabolizing enzymes, and genetic and environmental factors lead to the production of a range of enzymes that catalyze xenobiotic transformations, resulting in changes in P450 expression [43]. The Cytochrome P450 Family 1 Subfamily A Member 1 (CYP1A1) gene in this pathway showed an up-regulation trend, and CYP1A1 up-regulation was also found in *Takifugu rubripes* [44] and *Gillichthys mirabilis* [45]. We speculate that this may be related to the oxygen consumption capacity of *S. marmoratus*. In addition, ribosome biogenesis in eukaryotes was significantly down-regulated in the high-salt group. Ribosome biogenesis is the basis of cellular growth capacity and is inextricably linked to cellular processes such as growth, senescence, and cell division [46,47]. Cells can increase the efficiency of protein synthesis by producing new ribosomes to increase their translation capacity [48], so we hypothesize that the efficiency of protein synthesis in *S. marmoratus* is inhibited when the environment is too saline. When the salinity environment changes, more energy is required to maintain intracellular homeostasis and osmotic balance, and at this time, osmoregulatory tissues also provide a transient increase in energy, which helps to facilitate trans-epithelial ion transport [15,49]. Carbohydrate metabolism has been shown to respond to energy requirements associated with salinity stress [50], when salinity fluctuates, gills, kidneys, and other tissues will also rely on glucose to increase energy consumption to meet osmotic regulation [51], and sugar plays an osmotic protection role for Na+ ions [52]. Glycolysis/gluconeogenesis was up-regulated in the low-salt group and down-regulated in the high-salt group, and this pathway produces an ATP-reducing equivalent (NAD (P) H+) in response to salinity stress. We speculate that this is related to the salt tolerance mechanism of *S. marmoratus* and that the up-regulation of the relevant genes in the low-salt group may mean that the salt tolerance limit of *S. marmoratus* has been reached and that *S. marmoratus* needs to increase energy to maintain the osmotic balance between the internal and external environment. In the low-salt group, the up-regulated genes of *S. marmoratus* were significantly enriched in the cellular senescence pathway. Cellular senescence is a stress response caused by both intrinsic and extrinsic injury, and cellular senescence in fish cells increases cell membrane stiffness [53]. The expression of p38 MAP kinase, which is involved in physiological processes such as inflammation, cell death, and cellular senescence, is significantly increased in this pathway. It has been shown that p38 MAP kinase activity is significantly increased under hyperosmotic stress and decreased under hypoosmotic stress [54,55,56]. Thus, the increased expression of p38 MAP kinase may represent the cell death, senescence, and growth of *S. marmoratus* in a low-salt environment. In contrast, the high-salt group showed a significant decrease in the growth arrest and DNA damage-inducible protein GADD45 beta-like (Gadd45) gene, which is involved in physiological processes such as apoptosis and DNA damage, in the cellular senescence pathway [57]. Previous studies have shown an increased expression of Gadd45 in *Oncorhynchus kisutch* under high-salt conditions [45]. Cell proliferation induced by hormones and growth factors may play a role in osmoregulation, and DNA damage sensors have been suggested as possible osmotic sensors that recognize osmotic stress and initiate signal transduction pathways [16,58]. Thus, the inhibition of the cellular decay pathway in a high-salt environment may have affected the growth and development of *S. marmoratus.* In addition, the calcium signaling pathway was significantly down-regulated in the low-salt group, and this pathway is thought to play an important role in ion exchange and osmotic pressure regulation [52]. The activity of plasma membrane calcium-transporting ATPase (atp2a2 and atp2a3) ADP/ATP translocase (slc25a4 and slc25a5) was suppressed, and these genes may be involved in osmoregulatory processes. The ion transporter proteins (atp2a2 and atp2a3) are known to be related genes for hypotonic regulation in *Takifugu obscurus* [59], so we hypothesize that *S. marmoratus* relies on the switching of ion channels to regulate the osmotic pressure of the internal environment and ensure the isotonic state between the organism and the outside world when exposed to salinity stress and that calcium signaling pathway is closely related to this process.

In addition, three differential genes were significantly expressed in all three salinity groups—pik3r6b, cPLA2γ-like, and wsb1. Interestingly, all three genes were highly expressed in the high salinity group, with wsb1 being more expressed than pik3r6b and cPLA2γ-like. The E3 ubiquitin ligase, WSB1, belongs to the SOCS box, a cytokine signaling suppressor that regulates various biological processes, including DNA damage and repair, growth and development, immune regulation, etc. Previous studies have shown that decreased expression of the wsb1 gene inhibits the growth and development of zebrafish embryos [60,61]. WSB1 is also involved in neuroprotective pathways [62]. Fish can sense salinity fluctuations through neural signals, so *S. marmoratus* can protect neurons in response to osmotic stress. In addition, pik3r6b and cPLA2γ may cause organ dysfunction and cell damage in *S. marmoratus*. Phosphatidylinositol 3-kinases (PI3Ks) are lipid kinases and pik3r6b is the regulatory subunit of the PI3Kγ complex, which has been found to regulate the cardiovascular system [63,64]. When fish are subjected to salinity stress, changes in environmental salinity can lead to changes in body leucocyte counts, internal organs, and even death [65,66,67]. Cytoplasmic phospholipase A2γ (cPLA2γ) belongs to the cPLA2 (cytosolic phospholipase A2) family. cPLA2γ has structural expression in the endoplasmic reticulum. There is a structural expression in the endoplasmic reticulum, which also plays an important role in membrane phospholipid remodeling and maintenance, inflammation, and contractile dysfunction [68,69,70]. Excessive salinity can damage membrane systems, proteins, and nucleic acid molecules [71]. Overall, we speculate that fluctuations in salinity can cause changes in a range of physiological processes in *S. marmoratus* and even lead to mortality, all of which are related to the salt tolerance mechanism of *S. marmoratus*, but the exact regulatory mechanisms need to be further investigated.

## 5. Conclusions

Salinity is one of the environmental factors that fish depend on for survival, and as a euryhaline fish, it is essential to study the salinity tolerance mechanisms of *S. marmoratus*. In this study, RNA-seq analysis was performed on *S. marmoratus* exposed to water environments in the high-salt group (35‰) vs. the low-salt group (10‰), and the results showed that a total of 479 and 520 DEGs were obtained in gill tissues, and many genes were up-regulated in the high- and low-salt groups, and these genes may be related to osmoregulatory processes. Gene annotation and functional analysis revealed that osmoregulation in S. marmoratus is mainly regulated through carbohydrate metabolism, calcium signaling pathway, and glycolysis/gluconeogenesis. In addition, changes in salinity may have some effects on physiological processes, such as osmoregulation, growth and development, and cell proliferation in *S. marmoratus*. In summary, the results of the transcriptome analysis provide data and a theoretical basis for the adaptation mechanism of *S. marmoratus* to salinity changes. However, further studies on the regulatory mechanisms are needed.

## Figures and Tables

**Figure 1 animals-13-00400-f001:**
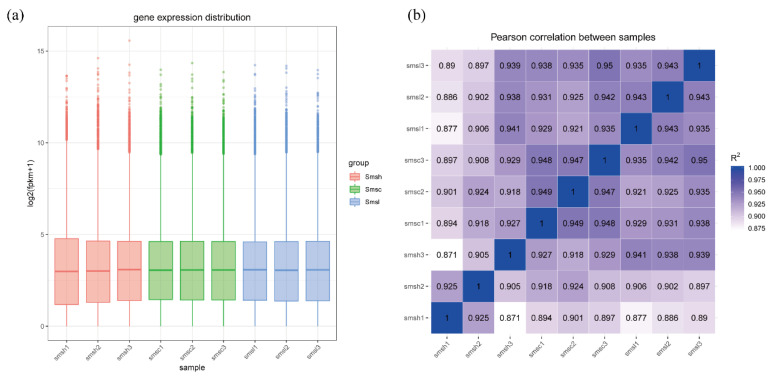
Expression level analysis diagram. (**a**) Gene expression box diagram (Appendix A: Gene_fpkm). (**b**) Sample correlation heatmap of gene expression (Appendix A: Gene_correlation).

**Figure 2 animals-13-00400-f002:**
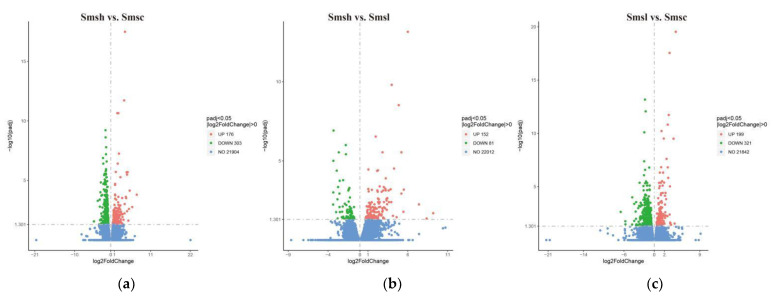
Volcano plot of DEGs in different salinity comparison groups. (**a**) Smsh vs. Smsc volcano plot; (**b**) Smsh vs. Smsl volcano plot; and (**c**) Smsl vs. Smsc volcano plot (Appendix A: Diff_stat).

**Figure 3 animals-13-00400-f003:**
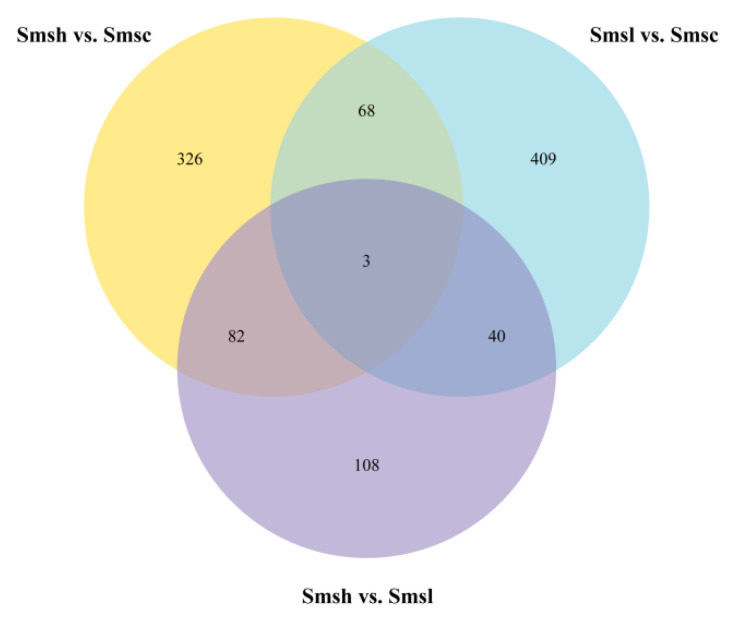
Venn diagram of differentially expressed genes.

**Figure 4 animals-13-00400-f004:**
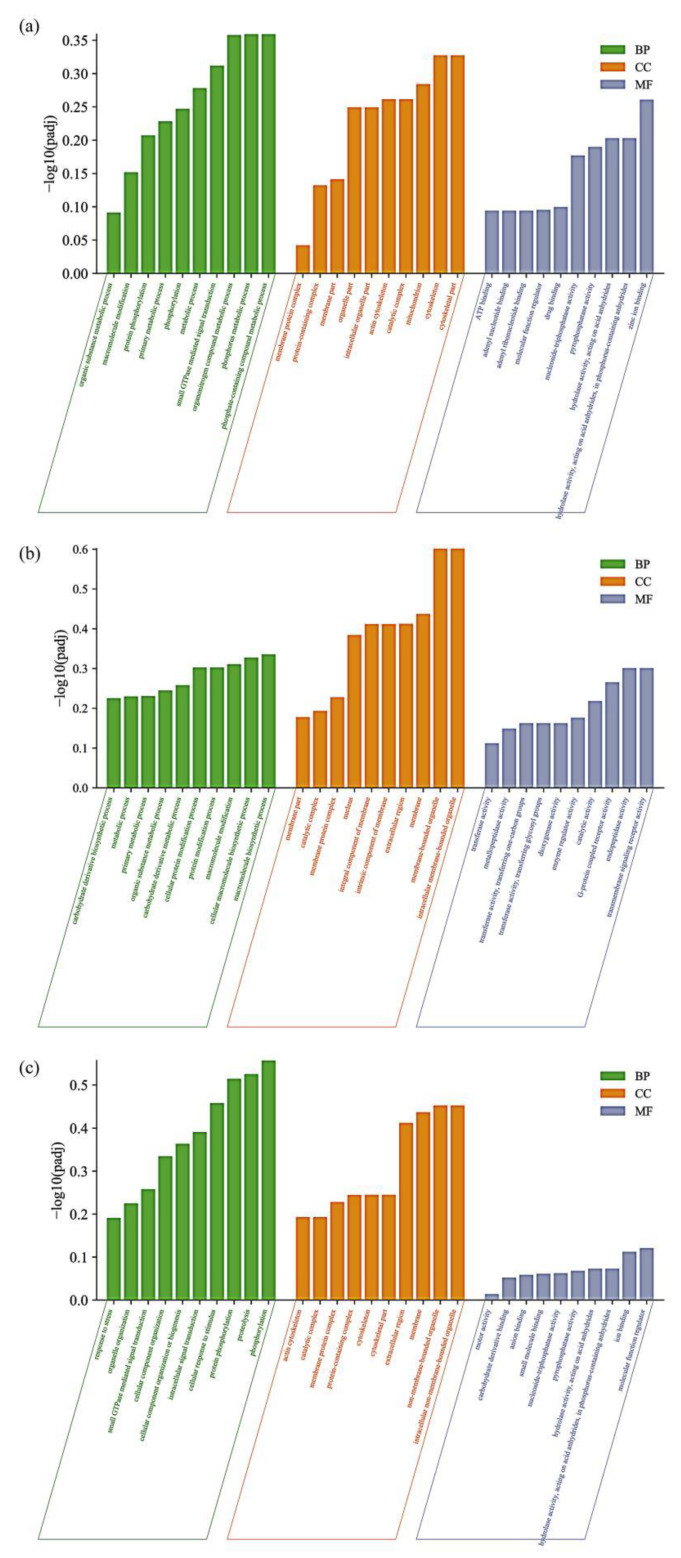
GO analysis of DEGs in different salinity comparison groups. (**a**) GO analysis of DEGs in the Smsh vs. Smsc; (**b**) GO analysis of DEGs in the Smsh vs. Smsl; and (**c**) GO analysis of DEGs in the Smsl vs. Smsc.

**Figure 5 animals-13-00400-f005:**
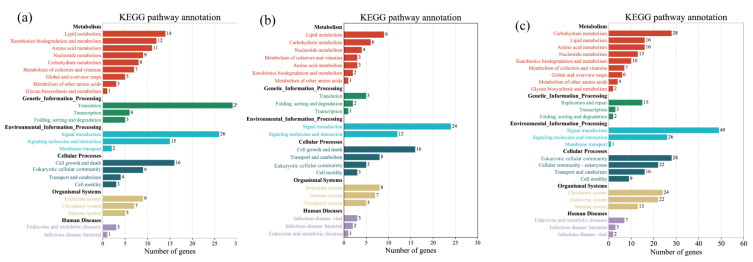
Annotation diagram of the KEGG pathway. (**a**) Annotated diagram of Smsh vs. Smsc KEGG pathway. (**b**) Annotated diagram of Smsh vs. Smsl KEGG pathway. (**c**) Annotated diagram of Smsl vs. Smsc KEGG pathway.

**Figure 6 animals-13-00400-f006:**
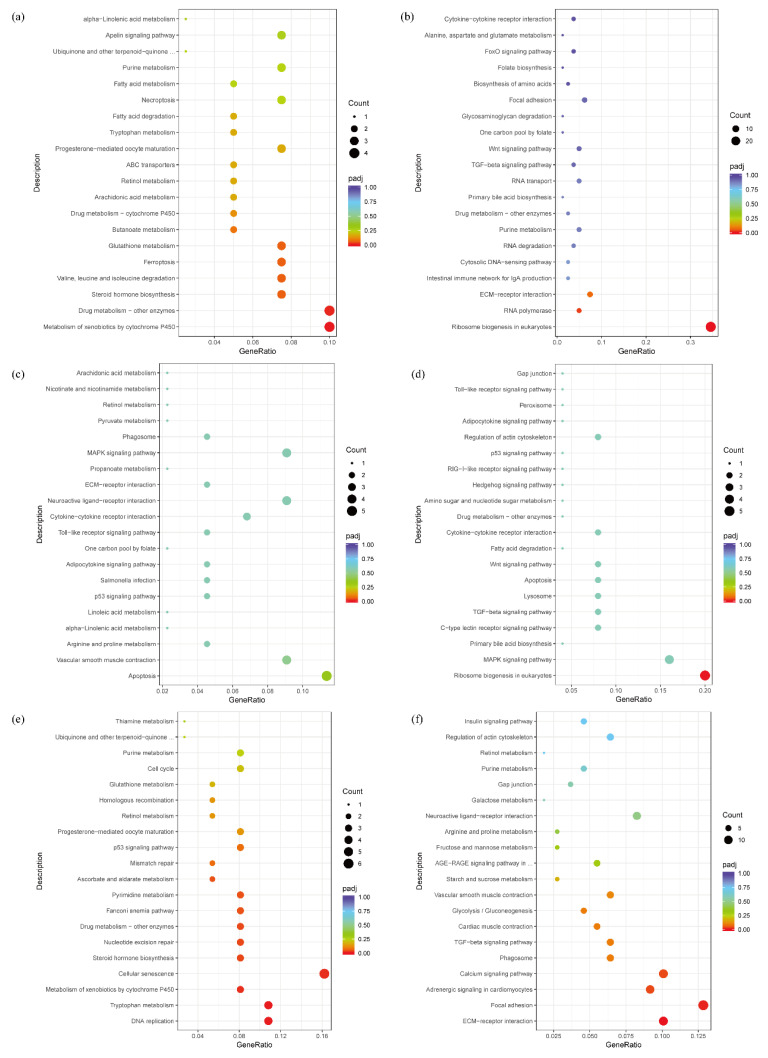
KEGG analysis of DEGs in different salinity comparison groups. (**a**) Smsh vs. Smsc dot plot of up-regulated DEGs; (**b**) Smsh vs. Smsc dot plot of down-regulated DEGs; (**c**) Smsh vs. Smsl dot plot of up-regulated DEGs; (**d**) Smsh vs. Smsl dot plot of down-regulated DEGs; (**e**) Smsl vs. Smsc dot plot of up-regulated DEGs; and (**f**) Smsl vs. Smsc dot plot of down-regulated DEGs.

**Figure 7 animals-13-00400-f007:**
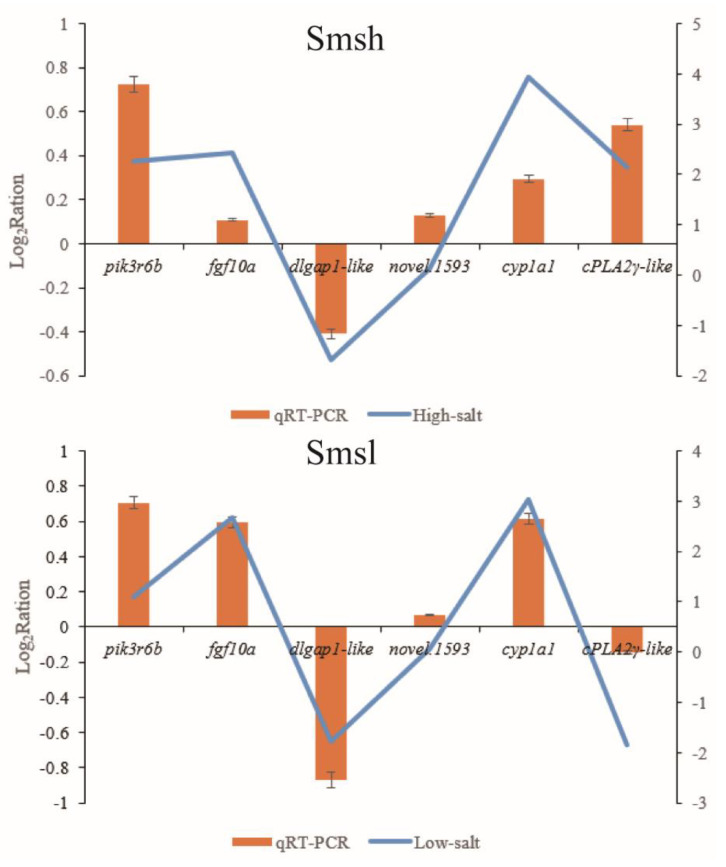
Validation of six significantly differentially expressed genes by qRT-PCR.

**Table 1 animals-13-00400-t001:** Primer sequences of differentially expressed genes under salinity stress were verified by qRT-PCR.

Gene	Primer Sequences (5′–3′)	Product Length
cPLA2γ-like	F: GCTGAGGTGCCGTCTTGTATC R: GGGAGGGTAGGGAGTGTTGAG	84
novel.1593	F: ACCGATCAACTGAAGAACCAA R: ATTATTGCGAGAACATCTGGAGG	85
cyp1a1	F: TGGCACCGAAGTCAACAAGC R: TGCTTCATTGTGAGACCGTATT	195
fgf10a	F: CCAACGGCAAGCCAATGAG R: CCAAAGAAGTCCGATACCCCG	143
pik3r6b	F: CCCAGAGTTCAGACACCTTA R: CTCCCTTTACAACTTCTCGT	140
dlgap1-like	F: CGTCAGCCGTCAATTTCCAGA	229
	R: CACCACCATTAACGCTCCCA	
β-actin	F: AGGGAAATCGTGCGTG R: ATGATGCTGTTGTAGGTGGT	233

**Table 2 animals-13-00400-t002:** Summary table of sequencing data quality.

Sample	Raw Reads	Clean Reads	Clean Bases	Error Rate	Q20	Q30	GC pct
Smsc1	55,156,824	54,422,314	8.16G	0.03%	97.06%	91.99%	48.64%
Smsc2	55,986,586	55,328,268	8.30G	0.03%	97.46%	93.04%	48.84%
Smsc3	48,404,256	47,888,456	7.18G	0.03%	97.30%	92.66%	48.80%
Smsh1	47,342,722	46,811,828	7.02G	0.03%	97.00%	92.04%	48.62%
Smsh2	52,238,798	51,609,812	7.74G	0.03%	97.21%	92.46%	48.62%
Smsh3	64,941,598	63,948,600	9.59G	0.03%	97.14%	92.36%	48.59%
Smsl1	56,895,700	56,030,266	8.40G	0.03%	97.51%	93.17%	49.18%
Smsl2	46,197,774	45,712,684	6.86G	0.03%	97.22%	92.50%	49.27%
Smsl3	59,582,084	58,994,516	8.85G	0.03%	97.59%	93.28%	48.65%

Note: Smsc—control group; Smsh—high-salt group; Smsl—low-salt group; Q20—the percentage of bases with a Phred value greater than 20 to the total bases; and Q30—the percentage of bases with a Phred value greater than 30 to the total bases; GCpct—the percentage of G and C of the four bases in clean reads.

**Table 3 animals-13-00400-t003:** Statistics of comparison between sample and reference genome.

Sample	Total Reads	Total Mapped	Unique Mapped	Multi Mapped	Positive Mapped	Negative Mapped
Smsc1	54,422,314	50,263,680(92.36%)	47,275,717(86.87%)	2,987,963(5.49%)	23,609,908(43.38%)	23,665,809(43.49%)
Smsc2	55,328,268	51,090,961(92.34%)	47,972,123(86.70%)	3,118,838(5.64%)	23,973,749(43.33%)	23,998,374(43.37%)
Smsc3	47,888,456	44,027,917(91.94%)	41,325,882(86.30%)	2,702,035(5.64%)	20,641,365(43.10%)	20,684,517(43.19%)
Smsh1	46,811,828	43,545,299(93.02%)	41,078,484(87.75%)	2,466,815(5.27%)	20,509,674(43.81%)	20,568,810(43.94%)
Smsh2	51,609,812	48,037,088(93.08%)	45,181,209(87.54%)	2,855,879(5.53%)	22,574,962(43.74%)	22,606,247(43.80%)
Smsh3	63,948,600	58,998,754(92.26%)	55,076,438(86.13%)	3,922,316(6.13%)	27,502,529(43.01%)	27,573,909(43.12%)
Smsl1	56,030,266	48,358,037(86.31%)	45,255,002(80.77%)	3,103,035(5.54%)	22,617,737(40.37%)	22,637,265(40.40%)
Smsl2	45,712,684	41,128,026(89.97%)	38,567,615(84.37%)	2,560,411(5.60%)	19,264,471(42.14%)	19,303,144(42.23%)
Smsl3	58,994,516	54,979,662(93.19%)	51,426,000(87.17%)	3,553,662(6.02%)	25,706,629(43.57%)	25,719,371(43.60%)

Note: Smsc—control group; Smsh—high-salt group; and Smsl—low-salt group.

**Table 4 animals-13-00400-t004:** The expression level of common genes in the transcriptome exposed to different salinity (FPKM).

Common DEGs	Expression Of Experimental Group		
	Smsc	Smsh	Smsl
pik3r6b	6.25	26.64	11.71
cPLA2γ-like	17.58	44.99	2.84
WSB1	89.76	350.29	73.19

## Data Availability

All relevant information is provided in this manuscript.

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
