# Peer review of "Transcriptome Analysis of Marbled Rockfish Sebastiscus marmoratus under Salinity Stress"

_animals, 2023, doi:10.3390/ani13030400_

Round 1

Reviewer 1 Report

In this study, RNA-seq sequencing was used to detect and analyze the differential gene expression of Sebastiscus marmoratus in salinity fluctuations, and to explore the genes and pathways related. However, even if the authors obtained some meaningful data, by and large, most of the results are provided by sequencing company and do not reflect the importance of the authors themselves. Moreover, this paper lacks innovation and the analysis of results is simple. I have raised some of these issues in my comments below, I hope the author will revise it carefully.

1.     Were the 30 fish temporarily reared before the experiment selected at random?

2.     What is the basis for 48 hours of temporarily kept before the experiment?

3.     What is the basis for choosing 12 h for the experiment? 

4.     The salinities in the experimental group are 10 and 35; is the salinity changing rapidly or gradually until it reaches the target salinity? If it is an acute change, then there is an ambiguity with the description according to L34-L37.

5.     L90-L91 This sentence should not appear in this position.

6.     Transcriptome sequencing has been widely used in the study of aquatic animals, and only the changes of gene expression can not clearly prove the physiological changes of aquatic animals. Therefore, more experimental parameters are needed, such as complement content, lysozyme activity, antioxidant enzyme activity….

Author Response

Response to Reviewer 1 Comments

Point 1: Were the 30 fish temporarily reared before the experiment selected at random?

Response 1: We thank the reviewer for pointing this out. In our study, before the experiment began, we randomly selected 30 fish of similar size in the offshore waters of Zhoushan. Thank you very much for your suggestion, it has been modified in L86.

Point 2: What is the basis for 48 hours of temporarily kept before the experiment?

Response 2: We thank the reviewers for asking this question. It has been shown that 48 h is the best time to keep fish in temporary kept [1]. The 48 hours of temporary kept before the experiment was to allow S . marmoratus to adapt to the laboratory environment, because the animals need to adapt to the new environment when moving from one environment to another, and also, during transportation and transfer, the animals' original rhythm will be affected and need to recover or adapt to the new rhythm. Therefore, in our study, we used the method of 48 hours of temporary rearing of fish.

Point 3: What is the basis for choosing 12 h for the experiment?

Response 3: We thank the reviewers for asking this question. Physiological processes are dynamic on hourly time scales and are often controlled by circadian rhythms, such as hormone secretion, drug and xenobiotic metabolism, and glucose homeostasis [2]. In previous studies, the experimental time was also set to 12 hours in order to reduce circadian-induced changes [3]. Therefore, in our study, the experimental time was set to 12 hours in order to reduce circadian-induced changes in S . marmoratus regulatory mechanism changes.

Point 4: The salinities in the experimental group are 10 and 35; is the salinity changing rapidly or gradually until it reaches the target salinity? If it is an acute change, then there is an ambiguity with the description according to L34-L37.

Response 4: Thank you very much for pointing out this error, it was acute salinity stress in this study and has been revised and added to the introduction as you suggested.

Point 5: L90-L91 This sentence should not appear in this position.

Response 5: According to reviewer suggestion. We have fixed this error in the manuscript.

Point 6: Transcriptome sequencing has been widely used in the study of aquatic animals, and only the changes of gene expression can not clearly prove the physiological changes of aquatic animals. Therefore, more experimental parameters are needed, such as complement content, lysozyme activity, antioxidant enzyme activity….

Response 6: We sincerely appreciate the valuable comments. It is very important, and thanks to your suggestions, we have identified shortcomings in our current work, It is true that our study only describes the changes in gene expression, as you mentioned. In the present study, we mainly focused on S. marmoratus under salinity fluctuations and explored the functional genes and pathways associated with salinity stress, and we think that focusing only on gene expression trends may not be optimal, but should be sufficient to conclude that S. marmoratus is involved in pathways related to immune and osmoregulation under salinity stress and that functional genes related to salinity have also been identified. Pathways such as the calcium signaling pathway, Glycolysis/gluconeogenesis, carbohydrate metabolism, cell growth and death, among which calcium signaling pathway and Glycolysis/gluconeogenesis are thought to produce important roles in ion exchange and osmotic pressure regulation. Our study elucidated key candidate genes involved in salinity regulation in S. marmoratus, and in subsequent studies, we further elucidated the physiological changes in salinity adaptation. We are very grateful for your suggestion, and we will continue to explore it in our future work according to your guidance in the follow-up study, further verify the conclusion with your suggested experimental parameters, and compare it with the current study.

Reference:

  • Nie, X.; Zhang, Y.; Sun, X.; Huang, B.; Zhang, C. Process and key technologies of transportation of live fish.FisheryModernization, 2014, 41, 34-39.

[2] Yeung J, Naef F. Rhythms of the genome: circadian dynamics from chromatin topology, tissue-specific gene expression, to behavior. TRENDS GENET, 2018, 34, 915-926.

[3] Lou F, Wang Y, Han Z, et al. Comparative transcriptome reveals the molecular regulation mechanism of Charybdis japonica to high-and low-temperature stresses. FRONT MAR SCI, 2022, 180.

Reviewer 2 Report

This manuscript entitled “Transcriptome analysis of marbled rockfish Sebastiscus marmoratus under salinity stress” by He et al. is well designed, and the results are of significance. It could be published in your journal after a small revision.

In Introduction section, the author needs to review more studies concerning the physiological changes and salinity stress, not only about fishes.

In Figure 7, the authors need to show what does the curve mean? Does it mean the results of transcriptome?

In Discussion section, the authors need to discuss it deeply, for example, what is the physiological effects of salinity stress, and what is the potential mechanism?

In Conclusion section, the authors need to simply conclude the results.

Author Response

Response to Reviewer 2 Comments

Point 1: In Introduction section, the author needs to review more studies concerning the physiological changes and salinity stress, not only about fishes.

Response 1: We sincerely appreciate the valuable comments. We have checked the literature carefully and added more references on physiological changes and salinity stress into the Introduction part in the revised manuscript. The relevant content is in L39-L43 and L45-L47.

Point 2: In Figure 7, the authors need to show what does the curve mean? Does it mean the results of transcriptome?

Response 2: Based on reviewer comment. The curve refers to the results of RNA-Seq, which we have modified in Section 3.5.

Point 3: In Discussion section, the authors need to discuss it deeply, for example, what is the physiological effects of salinity stress, and what is the potential mechanism?

Response 3: According to the reviewer's comment. We think this is an excellent suggestion. Salinity changes cause fish to dynamically regulate the osmotic pressure inside and outside the organism and cause changes in their feeding, metabolism, and enzymes, finally causing disturbances in the internal environment with abnormalities in physiological indicators. Therefore, we have added related content to L305-L311 and L320-L325.

Point 4: In Conclusion section, the authors need to simply conclude the results.

Response 4: Thank you for your suggestion, we have revised and added content to the conclusion section.

Reviewer 3 Report

The study examines the effect of a classic environmental stressor, i.e. salinity, in the transcriptome of S. marmoratus, a commercially important fish. The subject is interesting in the context of climate change effects on aquatic populations, as well as innovative. The manuscript is generally well written and the results very well discussed. I also find totally correct the fact that gene expression analyses were cross validated by real time PCR. 

However methodologically there are some points that not clarified and should be explained more. For instance firstly I could not understand how many samples were sequenced. All 6 from each treatment?

Secondly, why the 20‰ was chosen as the control group? Please provide a justification. Is it the normal salinity for wild populations? Maybe also give some references

Also some there is some missing information that should be added. Although S. marmoratus is eyryaline is there a known critical value under which it does not survive or threatened? Please add this info or at least any previous suggestions

Also some minor comments

In line 24 “were differentially expressed”, but also statistically significantly?

In line 53, after “Osteichthyes fish” the sentence should finish and then start a new one.

In section 2.2 please explain what “three sets” are consisted of. In total how many samples? Pools or each one separated? Also I do not totally agree with the term “RNA samples”, probably extracted or isolated RNA would be better

In Table 1, β-actin has no reverse primer, please correct

In line 231, should it be a comma after “(Figure 5)”?

Author Response

Response to Reviewer 3 Comments

Point 1: For instance firstly I could not understand how many samples were sequenced. All 6 from each treatment?

Response 1: Thank you very much for your suggestion, in this study we set up 3 biological replicates, 3 S. marmoratus in good condition were randomly selected for sampling in each of the 3 groups, and 3 samples were sequenced in each group, so there were 9 samples in total. This was done to eliminate intra-group errors and enhance the reliability of the results; the more biological replicates, the higher the true positive rate and the less influenced by the screening threshold, and the fewer missed differential genes.

Point 2: Why the 20‰ was chosen as the control group? Please provide a justification. Is it the normal salinity for wild populations? Maybe also give some references

Response 2: Based on the reviewer's comment. When collecting S. marmoratus, we first measured the natural seawater, and the result was 20‰, and previous studies showed that the optimal salinity range for brown redfish is 16-22[1], so we chose 20‰ as the control group.

Point 3: Also some there is some missing information that should be added. Although S. marmoratus is eyryaline is there a known critical value under which it does not survive or threatened? Please add this info or at least any previous suggestions

Response 3: As suggested by the reviewer, we have added more references to support this idea. The salinity range of the S. marmoratus is 10-34, and the optimum salinity range is 16-22. When the salinity is lower than 10 or higher than 34, it will affect the survival of the S. marmoratus and can lead to death [1]. We have added to L61-L63 based on your suggestion.

Point 4: In line 24 “were differentially expressed”, but also statistically significantly?

Response 4: We thank the reviewers for asking this question. In this study, the P values for pik3r6b, cPLA2γ-like, and WSB1 were <0.05, and because the smaller the P value the more significant the expression level of the differential genes, therefore, we considered the differences to be statistically significant.

Point 5: In line 53, after “Osteichthyes fish” the sentence should finish and then start a new one.

Response 5: Thank you very much for pointing out this error, we have fixed this error in the manuscript.

Point 6: In section 2.2 please explain what “three sets” are consisted of. In total how many samples? Pools or each one separated? Also I do not totally agree with the term “RNA samples”, probably extracted or isolated RNA would be better

Response 6: We thank the reviewers for asking this question. The "three sets" are three salinity groups, each with 3 samples, for a total of 9 samples, each separated from the other. Thank you for your suggestion, the use of "RNA samples" is not really appropriate, and the content of section 2.2 has been modified according to your suggestion.

Point 7: In Table 1, β-actin has no reverse primer, please correct

Response 7: Thank you very much for pointing out this error. In our resubmitted manuscript, the reverse primer for β-actin has been added to Table 1. Thank you for the correction.

Point 8: In line 231, should it be a comma after “(Figure 5)”?

Response 8: We thank the reviewer for pointing this out. Thank you for your reminder.

Reference:

[1] Wu C. The Effect of Several Environmental Factors on the Surival Rate of Larvae. Journal of Zhejiang Ocean University(Natural Science), 2000, 12-16.

Reviewer 4 Report

Comments:

This is an interesting paper (animals-2095810) providing an extensive “Transcriptome analysis of marbled rockfish Sebastiscus marmoratus under salinity stress”. However, I have major concerns about materials & methods (M&M) and results as mentioned below. 

Materials and Methods

Ln 89 – What kind of anesthesia was provided, and how much was given? State it!

Ln 116 – Is there no public database for reference genome of S. marmoratus? Note that for this kind of study, using a reference genome available in public databases is highly recommended!

Ln 136 - Did the authors conduct the melting curve analysis for the exclusion of primer

combinations forming primer/dimers and specificity confirmation of newly designed primers? State it in ms!

Ln140 - Detail procedure and program for qPCR should be explained here 

Ln 143- Include the references in order to support this statement!

Results

Ln 154 - The authors need to upload all raw and processed data to a respiratory, such as NCBI’s SRA database, before this manuscript will be accepted for publication and please state the accession number in the text.

Author Response

Response to Reviewer 4 Comments

Point 1: Ln 89 – What kind of anesthesia was provided, and how much was given? State it!

Response 1: We thank the reviewer for pointing this out. In this study, we used frost anesthetized, where gill tissue was removed after approximately 30 minutes of hypothermia at -20°C. Thank you for your suggestion and we have made changes in the manuscript in L97.

Point 2: Ln 116 – Is there no public database for reference genome of S. marmoratus? Note that for this kind of study, using a reference genome available in public databases is highly recommended!

Response 2: Thank you very much for the reminder that we have used the reference genome from the public database and noted it in section 2.3.

Point 3: Ln 136 - Did the authors conduct the melting curve analysis for the exclusion of primer

combinations forming primer/dimers and specificity confirmation of newly designed primers? State it in ms!

Response 3: We appreciate that you found this error. In this study, we performed the melting curve analysis and made changes and additions in section 2.5 of the manuscript.

Point 4: Ln140 - Detail procedure and program for qPCR should be explained here.

Response 4: We appreciate your suggestion, and we have made changes in section 2.5.

Point 5: Ln 143- Include the references in order to support this statement!

Response 5: Thank you for discovering this error, we have added the references in the manuscript (Ref. 34).

Point 6: Ln 154 - The authors need to upload all raw and processed data to a respiratory, such as NCBI’s SRA database, before this manuscript will be accepted for publication and please state the accession number in the text.

Response 6: We sincerely appreciate the reviewer's valuable comments. We obtained raw reads from 9 samples of S. marmoratus subjected to different salinity treatments by Illumina-based RNA sequencing and deposited them into the National Center for Biotechnology Information database and noted in L164-L165 of the manuscript.

Round 2

Reviewer 4 Report

Comments:

The manuscript (animals-2095810-v1) has been well-improved according to the previous comments and I would recommend it for publication.